# Glutaminases as a Novel Target for SDHB-Associated Pheochromocytomas/Paragangliomas

**DOI:** 10.3390/cancers12030599

**Published:** 2020-03-05

**Authors:** Balazs Sarkadi, Katalin Meszaros, Ildiko Krencz, Letizia Canu, Lilla Krokker, Sara Zakarias, Gabor Barna, Anna Sebestyen, Judit Papay, Zoltan Hujber, Henriett Butz, Otto Darvasi, Peter Igaz, Judit Doczi, Michaela Luconi, Christos Chinopoulos, Attila Patocs

**Affiliations:** 12nd Department of Internal Medicine, Semmelweis University, 1088 Budapest, Hungary; sarkadi.balazs2@med.semmelweis-univ.hu (B.S.); sarazakarias@gmail.com (S.Z.); igaz.peter@med.semmelweis-univ.hu (P.I.); 2Hereditary Tumours Research Group, Hungarian Academy of Sciences and Semmelweis University, 1085 Budapest, Hungary; kati.balla@gmail.com (K.M.); krokker.lilla@gmail.com (L.K.); butz.henriett@med.semmelweis-univ.hu (H.B.); otto.darvasi@gmail.com (O.D.); 3Department of Laboratory Medicine, Semmelweis University, 1089 Budapest, Hungary; 4Bionics Innovation Center, 1088 Budapest, Hungary; sebestyen.anna@med.semmelweis-univ.hu; 51st Department of Pathology and Experimental Cancer, Semmelweis University, 1085 Budapest, Hungary; krencz.ildiko@med.semmelweis-univ.hu (I.K.); barna.gabor@med.semmelweis-univ.hu (G.B.); papay.judit@med.semmelweis-univ.hu (J.P.); zoltan.hujber56@gmail.com (Z.H.); 6Department of Experimental and Clinical Biomedical Sciences “Mario Serio”, University of Florence, 50139 Florence, Italy; letizia.canu@unifi.it (L.C.); Luconi.michaela.luconi@unifi.it (M.L.); 7Department of Molecular Genetics, National Institute of Oncology, 1122 Budapest, Hungary; 8Molecular Medicine Research Group, Hungarian Academy of Sciences and Semmelweis University, 1085 Budapest, Hungary; 9Department of Medical Biochemistry, Semmelweis University, 1094 Budapest, Hungary; doczi.judit@med.semmelweis-univ.hu (J.D.); chinopoulos.christos@med.semmelweis-univ.hu (C.C.)

**Keywords:** succinate, SDH, SDHB, pheochromocytoma, paraganglioma, GLS-1

## Abstract

Pheochromocytoma/paragangliomas (Pheo/PGL) are rare endocrine cancers with strong genetic background. Mutations in the *SDHB* subunit of succinate dehydrogenase (SDH) predispose patients to malignant disease with limited therapeutic options and poor prognosis. Using a host of cellular and molecular biology techniques in 2D and 3D cell culture formats we show that SDH inhibition had cell line specific biological and biochemical consequences. Based on our studies performed on PC12 (rat chromaffin cell line), Hela (human cervix epithelial cell line), and H295R (human adrenocortical cell line) cells, we demonstrated that chromaffin cells were not affected negatively by the inhibition of SDH either by siRNA directed against *SDHB* or treatment with SDH inhibitors (itaconate and atpenin A5). Cell viability and intracellular metabolite measurements pointed to the cell line specific consequences of SDH impairment and to the importance of glutamate metabolism in chromaffin cells. A significant increase in glutaminase-1 (GLS-1) expression after SDH impairment was observed in PC12 cells. GLS-1 inhibitor BPTES was capable of significantly decreasing proliferation of SDH impaired PC12 cells. Glutaminase-1 and SDHB expressions were tested in 35 Pheo/PGL tumor tissues. Expression of GLS1 was higher in the SDHB low expressed group compared to SDHB high expressed tumors. Our data suggest that the SDH-associated malignant potential of Pheo/PGL is strongly dependent on GLS-1 expression and glutaminases may be novel targets for therapy.

## 1. Introduction

The unique metabolic environment of cancers is long known [1]. Pheochromocytomas/paragangliomas (Pheo/PGL) are rare (incidence: 0.8 per 100,000 person-years [2]) chromaffin cell derived neoplasms. In the past decade, enzymes of the tricarboxylic acid (TCA) cycle became the center of attention, because variants of genes encoding the subunits of succinate dehydrogenase enzyme [3,4,5,6], fumarate hydratase [7], malate dehydrogenase type 2 [8], and aspartate aminotransferase [9] enzymes have been associated with development of Pheo/PGL. The most widely accepted assumption is that defects of tricarboxylic acid (TCA) cycle may result in accumulation of certain, so called oncometabolites (such as succinate, fumarate, D-2-hydroxyglutarate [10]) which contribute to cancer development. Succinate competitively inhibits the 2-oxoglutarate (2-OG)-dependent HIF prolyl-hydroxylases in the cytosol [11,12], which results in stabilization and activation of HIF-1α therefore a shift to a pseudo-hypoxic environment occurs. This phenomenon is further demonstrated by the highly vascular phenotype of these tumors. In addition, chronic hypoxia (i.e., high altitudes) increases the incidence of sporadic PGLs and it has been demonstrated that it has a phenotype modifier effect on germline *SDHB* and *SDHD* mutant PGLs [13,14,15,16].

Even though germline mutations of genes encoding for *SDH* subunits have been shown to predispose susceptibility for the development of familial Pheo/PGL, only mutations of the *SDHB* gene have been often associated with high rate of malignancy. Metastatic disease can be observed in more than 17–40% of patients with *SDHB* mutations [17,18,19], but the mechanisms leading to the malignant phenotype are still unclear. The lack of a useful in vivo animal model for the development of Pheo/PGLs highly determines the experimental opportunities. [20]. Due to the lack of response to the currently available therapy for malignant Pheo/PGL, novel and easily accessible in vitro models for this tumor are required in order to evaluate the candidate therapies and to uncover new prognostic and therapeutic targets.

Glutamine is a major source of carbon for nucleotide and non-essential amino acid biosynthesis [21], and its metabolism supports cell proliferation [22]. Glutamine also serves as an energy source through glutamine-driven oxidative phosphorylation [23], as it replenishes TCA intermediates. SDHB-deficient cells show increased glutamine incorporation, which might be used as a shuttle for aspartate from the mitochondria to the cytosol to support cellular anabolism [24]. Glutamine metabolism also yields precursors for glutathione production, thus plays an important role in maintaining the redox homeostasis of cancer cells [25,26,27]. Furthermore, glutaminolysis supports substrate-level phosphorylation during hypoxia in tumors [28].

Located in the mitochondria, glutaminase-1 (GLS-1) generates glutamate from glutamine. Glutamate can be further metabolized to α-ketoglutarate, by glutamate dehydrogenase (GDH), which can directly fuel the TCA cycle. GLS-1 has been found to be upregulated in some cancers, and in some cases deregulated glutamine metabolism is essential for cancer growth [29,30,31,32]. *SDHx* mutant tumors were shown to accumulate lower levels of glutamate [33], and *SDHB* knockout cells were shown to be more sensitive to GLS-1 inhibitors [34]. Targeting glutamine metabolism in SDH deficient cancer is emerging as an ongoing trial (NCT02071862) including, inter alia, *SDH* associated gastrointestinal stromal tumors and non-gastrointestinal stromal tumors. However, to date, there are only very limited published data available about the efficacy of GLS-1 inhibitors in *SDHB* related malignancies [35].

Itaconate is a natural metabolite, in vivo it is synthesized in macrophages from cis-aconitate by cis-aconitase, coded by *Irg1* (immunoresponsive gene 1) in order to dysregulate bacterial metabolism [36]. Itaconate contributes to macrophages’ antimicrobial activity by inhibiting isocitrate lyase of bacteria [37,38] and to limit neuronal Zika virus infection by inducing an antiviral intracellular metabolic state [39]. Itaconate can reduce the activity of SDH in vitro [40] in a dose dependent manner, but has no effect on other mitochondrial pathways [41].

In addition, it was shown that itaconate can facilitate tumor progression through a ROS-driven pathway [42]. It was demonstrated that peritoneal tissue-resident macrophages promote tumor progression in certain tumors, including melanoma and ovarian carcinoma by tumor induced *Irg1* expression resulting in high itaconic acid levels. This pro-tumor effect was associated with the reactive oxygen species mediated MAPK activation in tumor cells [43], to the best of our knowledge, there are no data examining the effects of itaconate on cell survival.

Atpenin A5 (atpenin) is an SDH inhibitor that binds in the ubiquinone binding pocket comprised of residues from SDH subunits B, C, and D, blocking the electron transfer between the enzyme and ubiquinone [44,45]. It is important to note that the inhibition of SDH with atpenin could not induce hypoxia mediated gene expression in monocytes [46] and a dose dependent reduction of cell survival after treatment with atpenin analogues has been shown [47].

In this current work we aimed to study the biological and metabolic consequences of accumulation of succinate obtained through pharmacological and translational inhibition of the SDH enzyme in various cancer cell lines and using siRNA knockdown of *Sdhb* in rat pheochromocytoma cell line, PC12. Our complex in vitro study revealed that SDH inhibition facilitated the viability of chromaffin cells but not the non-chromaffin cells. Selective inhibition of GLS-1 enzyme decreased the proliferation of SDH impaired PC12 cells in monolayer and in 3D tissue culturing. Based on our in vitro findings, we detected an upregulation of GLS-1 in *SDHB*-low expressed Pheo/PGL tumors compared to *SDHB* highly expressed Pheo/PGLs. Our data pointed to the importance of the choice of cell line for studying SDH impairment and indicated the potential prognostic role and therapeutic target of GLS-1 enzyme in *SDH*-associated malignant Pheo/PGL.

## 2. Results

### 2.1. Sdhb Targeting siRNA Effectively Decreased SDH Activity

PC12 cells were transfected with two different *Sdhb* targeting siRNAs. After 48 h incubation, SDH activity SDHB protein levels and succinate/fumarate ratios were assessed. SDH activity was effectively reduced after si*Sdhb* transfection compared to mock transfected and untreated cells (Figure 1A–D). si*Sdhb* transfection showed similar potential in inhibiting SDH activity to atpenin, a potent and known SDH inhibitor (Figure 1D). Combination of two *SDHB* targeting siRNAs effectively reduced SDHB protein levels (Figure 1E,F and Appendix A). Succinate/fumarate ratio increased significantly in cells transfected with siRNA against *Sdhb* compared to cells transfected with mock siRNA (*p* < 0.0001) (Figure 1G).

### 2.2. Itaconic Acid Treatment Successfully Inhibited SDH Activity in All Cell Lines Studied

Succinate/fumarate ratio significantly increased in PC12 cells after 24 h (*p* < 0.0001) and 48 h (*p* < 0.0001) itaconate treatment. (Appendix A)

Similar to PC12 cells, significant increase in succinate/fumarate ratio was observed in HeLa cells after 24 h (*p* < 0.0001) and 48 h (*p* < 0.0001) itaconate treatment, as well as in H295R cells after 24 h (*p* < 0.0001) and 48h (*p* < 0.0001) itaconate treatment. (Appendix A)

### 2.3. Atpenin Treatment Successfully Inhibited SDH Activity in All Cell Lines Studied

Succinate/fumarate ratios due to the immeasurable concentrations of fumarate in case of atpenin treatment could not be calculated. However, SDH activity was also successfully inhibited by atpenin treatment based on the significant (*p* < 0.0001) increase in succinate concentrations and the significant (*p* < 0.0001) fold increase in succinate levels in all cell lines, compared to control (Appendix A).

### 2.4. Cell Viability and Proliferation

#### 2.4.1. SDH Impairment Had an Overall Positive Effect on Cell Viability Without Significant Changes in the Proliferation in PC12 Cells

*Sdhb* knockdown significantly increased PC12 cells’ viability after 72 h (*p* = 0.04) compared to mock transfected cells whereas significant differences were not observed at 24 and 48 h. (Figure 2A). A significant difference was observed in itaconic acid treated PC12 cells compared to vehicle treated cells after 24 h (*p =* 0.026) but not at 48 and 72 h (Figure 2B). Atpenin treatment yielded an increase in cell viability, but not at a significant level in the PC12 cell line (Figure 2C).

Cell proliferation of PC12 cells measured with SRB assay was not affected by SDH impairment either with *Sdhb* knockdown or itaconate/atpenin treatment (Figure 2D–F).

#### 2.4.2. Itaconate Decreased Cell Viability of HeLa and H295R Cells Whereas Atpenin Only Decreased Cell Viability in the H295R Cell Line

HeLa cells showed significant decrease in cell viability upon 48 h (*p =* 0.002) and 72 h (*p =* 0.002) treatment with itaconate. The opposite effect was observed after atpenin treatment of HeLa cells with significant increase in cell viability at 48 h (*p =* 0.015) compared to vehicle treatment (Figure 3A,B).

The H295R cell line showed an overall negative response to SDH impairment: significant decrease in cell viability was observed upon itaconate treatment after 72 h (*p =* 0.0043) and upon atpenin treatment after 24 h (*p =* 0.004) and 48 h (*p =* 0.017) (Figure 3C,D).

#### 2.4.3. Changes in Glutamate and Lactate Concentrations were SDH Inhibition Method and Cell Line Specific

The metabolite concentrations for each cell line and each SDH inhibitory method are presented in Figure 4 and Appendix A, and Appendix A.

In PC12 cells metabolite concentrations measured after *Sdhb* knockdown clustered together to those measured after itaconate treatment (Figure 4B). Succinate accumulation was detected in all cell lines after inhibition of SDH activity. *Sdhb* knockdown significantly decreased glutamate concentrations (*p* < 0.0001) (Figure 4C), while lactate did not show accumulation (Figure 4D and Appendix A).

Glutamate levels also decreased in PC12 cells after itaconic acid treatment compared to vehicle treatment after 24 h (*p* = 0.008) and 48 h (*p* = 0.53) without lactate accumulation (Figure 4C,D and Appendix A). Atpenin induced a significant decrease of glutamate concentrations in PC12 cells after 24 h (*p* < 0.0001) and 48 h (*p* < 0.0001), accompanied with significant increases in lactate concentrations after 24 h (*p* = 0.004) and 48 h (*p* < 0.0001) (Figure 4C,D and Appendix A).

Contrary to PC12 cells, glutamate (*p* = 0.004) and lactate (*p* = 0.013) concentrations significantly increased in HeLa cells treated with itaconic acid compared to vehicle treatment after 24 h incubation. Elevation of both glutamate (*p* < 0.0001) and lactate (*p* = 0.018) concentrations were also significant after 48 h incubation (Appendix A and Appendix A).

Atpenin treatment resulted in a significant decrease in the glutamate concentrations of HeLa cells after 24 h (*p* < 0.0001) and 48 h (*p* < 0.0001) accompanied by significant increases in lactate concentrations after 24 h (*p* = 0.0039) and 48 h (*p* < 0.0001) (Appendix A and Appendix A).

In H295R cells itaconic acid treatment caused significant increase in intracellular glutamate (*p* = 0.018) and lactate (*p* < 0.0001) levels only after 24 h (Appendix A and Appendix A). Glutamate concentrations significantly decreased in H295R cells after 24 h (*p* < 0.0001) and 48 h (*p* < 0.0001) atpenin treatment accompanied by significant increases in lactate concentrations after 24 h (*p* < 0.0001) and 48 h (*p* = 0.004) (Appendix A and Appendix A).

#### 2.4.4. GLS-1 Gene Expression was Cell Line and SDH Inhibitory Method Dependent

Based on the metabolomics and cell viability measurements we sought to assess the importance of glutamine/glutamate metabolism, especially the mitochondrial uptake of glutamine by glutaminase-1 (GLS-1). A significant increase in GLS-1 expression after *Sdhb* knockout (fold change: 1.53 ± 0.3, *p* = 0.002) was observed in PC12 cells A significant increase in GLS-1 expression was observed after itaconate treatment of PC12 cells after 24 h (fold change: 1.2 ± 0.03, *p* = 0.015) and 48 h (fold change 1.48 ± 0.13, *p* = 0.002). On the contrary, GLS-1 expression decreased after atpenin treatment in PC12 cells after 24 h (fold change: 0.89 ± 0.1, *p* = 0.065) and significant decrease was observed after 48 h (fold change: 0.82 ± 0.09, *p* = 0.002) (Figure 5A).

HeLa cells expressed a similar phenotype upon itaconate treatment: significant increase in GLS-1 expression was observed after 24 h (fold change: 1.47 ± 0.3, *p* = 0.002) and 48 h (fold change: 1.9 ± 0.63, *p* = 0.015) treatments. On the other hand, GLS-1 expression significantly decreased after 24 h atpenin treatment in HeLa cells (fold change: 0.47 ± 0.08, *p* = 0.002), but significantly increased after 48 h (fold change: 1.8 ± 0.37, *p* = 0.002) (Figure 5B).

H295R cells exhibited also a significant increase in GLS-1 expression after 24 h (fold change: 1.34 ± 0.1, *p* = 0.0022) and 48 h (fold change: 1.19 ± 0.1, *p* = 0.0152) itaconate treatment. Atpenin significantly increased GLS-1 expression both at 24 h (fold change: 1.78 ± 0.2, *p* = 0.0022) and 48 h (1.95 ± 0.2, *p* = 0.0022) (Figure 5C).

#### 2.4.5. Immunohistochemistry of SDHB and GLS-1 in Pheo/PGL Tissues Points to the Importance of GLS-1 Enzyme

Based on our in vitro findings we sought to evaluate the expression level of GLS-1 in Pheo/PGLs tumor tissues in order to assess whether GLS-1 expression might serve as a marker for malignancy in Pheo/PGLs.

Low (H-score < 100) SDHB staining characteristic for SDH-associated tumors was confirmed in all *SDHB*-associated tumor tissues while a high (H-score ≥ 100) staining was observed in *RET*-mutant tumors. Both high and low SDHB staining scores were observed in the sporadic tumor group (Table 1 and Figure 6).

Increased GLS-1 expression was detected in *SDHB*-mutant tumor tissues compared to *RET*-mutant and sporadic tumors, however the difference was not significant (H-score: 87.8 ± 64 vs. 59 ± 82.4 *SDHB*-mutant vs. *RET*-mutant, *p* = 0.22; H-score: 87.8 ± 26 vs. 53.6 ± 47.8, *SDHB*-mutant versus sporadic, *p* = 0.15). A total of 54% (7 of 13) of the low SDHB expressing tumors showed high GLS-1 staining while only 22% (five of 22) of high SDHB expressing tumors showed high GLS-1 staining (*p* = 0.07).

GLS-1 was overexpressed in three *RET*-mutant samples. Of these three samples, in two cases malignancy was proved as they were reoccurring, invasive, and metastatic Pheos of Patient No. 4 (Figure 7) and Patient No. 5. The third GLS-1 overexpressing *RET*-mutant Pheo sample was obtained from a patient with MEN2A syndrome with bilateral Pheo (Patient No. 6, the GLS-1 positive sample was the Pheo removed from right side).

In the three malignant sporadic Pheo/PGL samples, two showed high SDHB staining scores. All malignant sporadic samples showed low or average GLS-1 immunostaining. In case of the benign sporadic samples, out of the four samples with low SDHB scores two were accompanied by high GLS-1 immunostaining scores (Table 1).

#### 2.4.6. GLS-1 Inhibition in PC12 Cells Decreased Proliferation after SDH Inhibition Measured by SRB Assay

The proliferation of PC12 cells was not significantly different upon SDH impairment, compared to controls (Figure 2D–F).

To test whether SDH impaired PC12 cells’ proliferation is dependent on GLS-1 activity, we assessed the proliferation of the cells after BPTES treatment (BPTES is a selective GLS-1 inhibitor). Proliferation of PC12 cells significantly decreased when SDH inhibition was accompanied with BPTES treatment regardless of inhibitory methods (siRNA against *Sdhb* silenced cells after 72 h: *p* = 0.009; itaconate after 48 h: *p* = 0.009 and 72 h: *p* = 0.009; atpenin after 48 h: *p* = 0.009 and 72 h *p* = 0.002 (Figure 8A–C).

#### 2.4.7. GLS-1 Inhibition in Itaconate Treated 3D Cultured PC12 Cells Increased the Number of Dead Cells Compared to Vehicle Treated Cells

In order to assess GLS-1 inhibition in a more relevant in vitro model, we applied 3D culturing of PC12 cells by spheroid induction using spheroid inducing media. Itaconate treatment alone did not exert a significant effect on the ratio of living cells (compared to vehicle treatment 3% and 9% after 48 and 72 h treatment, respectively). When itaconate was accompanied by BPTES treatment in the 3D cultured PC12 cells, 18%, 13%, and 18% decreases were observed in the living cell ratios compared to vehicle treatment at 24, 48, and 72 h, respectively (*p* < 0.0001 for each comparison) (Figure 8D).

#### 2.4.8. Oxygen Consumption Rate Measurements

After biochemical characterization of SDH inhibition we assessed the mitochondrial respiration upon SDH inhibition using SeaHorse measurements in PC12 cells (Figure 9A). The effects of *Sdhb* knockdown were compared to mock transfected cells whereas the consequences of itaconate and atpenin treatment were compared to control (untreated) PC12 cells.

Basal respiration is derived by the subtraction of non-mitochondrial respiration from the baseline respiration. BPTES treated control cells yielded the lowest basal oxygen consumption ratio (OCR), whereas itaconate the highest. Compared to control PC12 cells, itaconate yielded a significantly higher basal respiration (*p* = 0.007) whereas *Sdhb* knockdown resulted in significantly lower basal respiration rate compared to mock transfected cells (*p* = 0.0079). BPTES treatment of cells transfected with siRNA against *Sdhb* or mock transfection did not result in a significant difference in OCR values (Figure 9B).

Basal respiration was then evaluated after administration of 2mM glutamine. Only minor changes were observed in itaconate (1.4%), atpenin (1.7%), and *Sdhb* silenced cells (0.5%). BPTES treatment only had a significant effect on *Sdhb* silenced cells when their basal oxygen consumption was compared to the OCR after glutamine admission (*p* = 0.0079). Similarly, a significant difference (*p* = 0.0079) was observed when the OCR of *Sdhb* silenced cells were compared to mock transfected cells after glutamine admission. BPTES treated *Sdhb* silenced and mock transfected cells’ OCRs after glutamine admission did not differ significantly (*p* = 0.15).

Maximal respiration is defined as the difference of OCR after 2,4-dinitrophenol (DNP) and after antimycin A + rotetone (A+R) admission. Itaconate treatment and si*Sdhb* knockdown significantly increased maximal respiration (*p* = 0.0079). BPTES treatment significantly reduced the maximal OCR of both control (*p* = 0.0079) and si*Sdhb* silenced (*p* = 0.0079) PC12 cells (Figure 9C).

Non-mitochondrial respiration is displayed after inhibition both of complex I and complex III with A+R. PC12 cells transfected with siRNA against *Sdhb* had the highest non-mitochondrial respiration which did not decrease significantly after BPTES treatment. Both itaconate (*p* = 0.0159) and si*Sdhb* treatment (*p* = 0.0079) significantly increased the non-mitochondrial respiration of PC12 cells (Figure 9D).

## 3. Discussion

Pheo/PGLs present a genetically heterogenic tumor group, arising from the adrenal medulla or the extra-adrenal paraganglia. A total of 40% of these neuro-endocrine tumors are inherited in an autosomal dominant manner due to mutations in one of the 17 Pheo/PGL-associated genes [48,49]. Of these genes, seven (*SDHA, SHDAF2, SDHB, SDHC, SDHD, FH, MDH2*) encode enzymes participating in the TCA cycle. Mutations of the *SDHB* gene represent a strong susceptibility for malignancy [50,51,52,53]. The precise pathomechanism behind the *SDHx* mutations and especially the malignant potential of *SDHB* mutations is still unknown despite the several observations made through the last decades [54,55,56]. Unfortunately, there is no therapeutic option for malignant cases which warrants further studies to identify novel therapeutic targets. Several novel approaches were introduced recently to address the lack of therapeutic options: the inhibition of glutathione synthesis was shown to contribute to the DNA damage as a result of the increased level of reactive oxygen species in *SDHB* mutant tumors [57]. Inhibition of complex I made complex II impaired tumors more sensitive to DNA damaging chemotherapeutic agents [58] while it has been also demonstrated that elevated succinate and fumarate levels suppress the homologous recombination DNA pathway, rendering these tumors vulnerable to poly(ADP)-ribose polymerase inhibitors [59]. In addition to the lack of therapeutic options, prognostic factors for the prediction of malignant disease are also mandatory for establishing a proper strategy for the management of the disease. However, previous attempts show that creating a universal prognostic factor for all etiologies of Pheo/PGL is hardly possible [60].

The succinate accumulation in *SDHx* mutant tumors can inhibit the α-ketoglutarate-dependent prolyl hydroxylases, which have an important role in the degradation of HIF1α and HIF2α under normoxia [12]. Mutations in the *SDHB* subunit beside the HIF1α stabilization, shift the cellular metabolism towards reductive glutamine catabolism [61]. Recently, Lorendeau et al. reported that both loss of complex I and complex II activity are necessary to mimic the metabolic phenotype of *SDH* mutant tumors based on reductive glutamine metabolism, sole *SDHA* or *SDHB* inhibition failed to do so in their study [62]. Our aim was to assess the consequences of SDH impairment in various cell types and to search for novel in vitro models, prognostic markers, and therapeutic targets for tumors with reduced or absent SDH activity.

Knockdown of *Sdhb* with siRNA in PC12 rat chromaffin cells successfully inhibited SDH activity and increased succinate/fumarate ratio by >3 fold compared to mock siRNA transfected cells. Increased succinate to fumarate ratio also characteristic for SDH mutant Pheo/PGLs [63]. Based on the metabolite measurements, both itaconate and atpenin were more potent SDH inhibitors than *Sdhb* knockdown.

Based on the cell viability and oxygen consumption measurements, PC12 cells were not affected negatively by SDH impairment. Moreover, these cells showed an overall positive response to SDH impairment while HeLa and H295R cells showed decreased viability after itaconate treatment. Atpenin also increased HeLa cell line viability, whereas decreased H295R cell line viability. Based on these data we assume that the impairment of SDH activity (either by itaconate or atpenin treatments or *Sdhb* knockdown) has a cell type-specific effect on the viability of cells. Significant difference in PC12 cells’ proliferation was not observed after SDH impairment in monolayer cell culture. Itaconate treatment of the 3D PC12 cell culture model did not decrease the ratio of living cells significantly. Based on these results we conclude that PC12 cells can cope with SDH impairment both in the monolayer and the 3D cell culture model.

Next, we sought to evaluate whether the cell viability effects can be traced back to the differences in the metabolite profiles observed after SDH inhibition (beside the differences in the succinate/fumarate ratios). In general, inhibition of SDH shifts cellular metabolism to anaerobic glycolysis, and administration of itaconate is also associated with lactate accumulation [64]. However, in contrary to HeLa and H295R cell lines, the expected increase in lactate concentrations was absent in PC12 cells after itaconate treatment and after *Sdhb* knockout. *Sdhb* knockdown significantly decreased glutamate concentrations which is in line with the data demonstrating that *SDHx* mutant tumors also accumulate lower levels of glutamate [33], and *SDHB* mutation associated with increased glutamine metabolism [35].

In addition, glutamine was shown to be the main source in SDHB-mutated UOK269 cells and this metabolite linked HIF-1α stabilization and DNA methylator phenotype [61].

Pursuing the role of glutamine in SDH impaired cells we studied the respiration of SDH impaired PC12 cells. These cells in the presence of glutamine effectively switch from glycolysis to glutaminolysis which increases the basal OCR values. These results are in line with the data published by Zhdanov et al., who showed that increase in the OCR values upon mitochondrial uncoupling was only seen when glutamine was combined with either glucose or pyruvate. In addition, the cell-specific dependence on glutaminolysis was also highlighted [23]. Itaconate but not atpenin had the same effect, it increased the basal respiration of PC12 cells whereas it did not have a significant impact on lactate concentration further supporting its capability to serve as a model for *Sdhb* mutant Pheo/PGLs.

The most significant effect related to oxygen consumption was observed in the non-mitochondrial respiration fraction, suggesting that these cells use non-mitochondrial respiration for survival. In pheochromocytoma and paragangliomas there is no data about the expression and role of mitochondrial uncoupling protein 2 (UCP2) which has been suggested to be a metabolic sensor of cells under nutrient shortage. We may hypothesize that in SDH deficient cells a rapid metabolic adaptation occurs which allows these cells to survive by either shifting its metabolism to the use of the alternative fuel glutamine or going into a reversible, more quiescent state [65].

Glutamate has an extensive role in cell metabolism [66] and disruption of the TCA cycle makes the cells more dependent on reductive carboxylation of glutamine instead of the oxidative metabolism of the TCA cycle [67,68,69]. GLS-1 is a mitochondrial enzyme that generates glutamate from glutamine, which further metabolizes to aspartate and α-ketoglutarate in the mitochondria. GLS-1 has been found to be upregulated in some cancers, and in some cases deregulated glutamine metabolism is essential for cancer growth [29,30,31,32].

Therefore, we sought to assess the expression of GLS-1 in vitro after SDH impairment. PC12 cells exhibited significantly increased GLS-1 expression upon *Sdhb* knockdown and SDH inhibition with itaconate too. Interestingly, atpenin treatment decreased the expression of GLS-1 in PC12 cells. HeLa cells also exhibited a significant increase in GLS-1 expression upon itaconate treatment. Similar to the PC12 cell line, atpenin treatment resulted in significantly decreased GLS-1 expression after 24 h in HeLa cells. However, this was reversed after 48 h, when a significant increase in GLS-1 expression was observed. The H295R cell line also displayed significantly increased GLS-1 expression after SDH inhibition by either itaconate or atpenin. These results indicate that SDH inhibition exhibits cell line and inhibitory method specific consequences and the dynamism of metabolic changes varies among cell types, but in PC12 cells both *Sdhb* knockdown and itaconate treatment increased its expression suggesting that these cells might be dependent on this enzyme. Contrary, in HeLa and H295R cells, increased GLS-1 expression was not necessarily associated with decreased glutamate concentrations. We hypothesize that even though the entry for glutamate is enhanced by the increased GLS-1 expression, glutamate is not used effectively after itaconate treatment in these cells which further emphasizes the importance of appropriate selection of in vitro models. In addition, further studies are warranted to clarify the role of GLS-1 in these cancer cells.

Based on these observations we evaluated the dependence on GLS-1 function of PC12 cells with impaired SDH activity. When SDH inhibition was accompanied by selective GLS-1 inhibition, PC12 cells showed significantly decreased proliferation in monolayer cell culture. Increased cell death was observed in the 3D PC12 cell culture model, suggesting that chromaffin cells with SDH impairment are dependent on the GLS-1 enzyme. It has to be mentioned that currently there is an ongoing clinical trial with the GLS-1 inhibitor CB-839 for *SDH*-associated gastrointestinal stromal tumors and non-gastrointestinal stromal tumors. However, an earlier study performed in pancreatic cancer showed the limited clinical efficacy of CB-839 monotherapy [70] which highlights again that various GLS-1 inhibitors may cause significantly different effects on chromaffin cells’ proliferation.

In order to translate our in vitro data to clinics we examined for the first time the expression of GLS-1 in various Pheo/PGL tumor tissues with known genetic background by immunohistochemistry. In line with in vitro data our immunohistochemistry analysis demonstrated an increased GLS-1 staining in *SDHB*-low expressed tumors compared to tumors with intact SDHB protein. Furthermore, a significant proportion of *SDHB*- and *RET*-associated malignant tumors also showed an increase in GLS-1 staining compared to benign *RET*-associated and sporadic tumors. It should be also mentioned that in some cases increased GLS-1 expression was not associated with malignancy. On the other hand, determination of malignancy in Pheo is difficult, because there is no obvious marker for it. Several studies, including a study published by Stenman et al., showed that even in *RET*-associated Pheos, using the “Pheochromocytoma of the Adrenal Gland Scaled Score “(PASS) and “Grading System for Adrenal Pheochromocytoma and Paraganglioma” (GAPP) algorithms, the malignancy was over-diagnosed [60]. In our study we considered a Pheo malignant when the tumor was recurring, or local or distal metastases were detected. In our in vitro experiments, the increased GLS-1 expression was not necessarily associated with increased viability, suggesting that for increased proliferation, other factors are also needed. The importance of GLS-1 may be the most important in SDH-compromised cells, where the concomitant inhibition of SDH and GLS-1 could result in cell lethality. The heterogeneous phenotype associated with *Sdhb* mutations is highlighted in an in vivo model of *Sdhb* mutation developed in *Caenorhabditis elegans*, where the deleted mutant arrested in development, while the point mutant form was viable and it presented only infertility [71]. This further supports personalized and case specific treatment of the disease.

In conclusion, we assume that GLS-1 contributes to *SDHB*-mutant malignant tumor growth and we presume that the evaluation of GLS-1 expression before therapy might yield valuable information for the management of the disease. A larger study evaluating malignant and benign Pheo/PGLs with various genetic backgrounds would clarify this observation and would decipher to role of GLS-1 in Pheo/PGL cells.

## 4. Materials and Methods

All materials were purchased from Merck-Sigma-Aldrich (Darmstadt, Germany), except where it is indicated in the text.

### 4.1. Cell Lines

All cell lines were obtained from American Type Culture Collection (ATCC). Cell cultures were incubated at 37 °C in a humidified 5% CO_2_/95% air atmosphere.

PC12 cells (rat pheochromocytoma cell line) were grown in 75-cm^2^ flasks in F-12 (# 21127022 F-12 Kaigh’s modification, Gibco, Thermo Fisher Scientific, Waltham, MA, USA), containing 15% horse serum (Gibco BRL), 5% fetal bovine serum (Gibco BRL), and 1% penicillin-streptomycin (Biosera LM-A4118/100). Culture media was replaced three times a week. Cells were removed from flasks for subculture and for plating into assay dishes using Trypsin-EDTA solution.

HeLa cells (human cervix carcinoma cell line) were grown in 75-cm^2^ flasks in Dulbecco’s modified Eagle medium/HamF12 (DMEM/F12) (#11330032, Thermo Fisher Scientific, Waltham, MA, USA) containing 10% FBS (#10270106, Thermo Fisher Scientific, Waltham, MA, USA) and 1% penicillin-streptomycin (LM-A4118/100, Biosera, Nouille, France). Culture media was replaced three times a week. Cells were removed from flasks for subculture and for plating into assay dishes using Trypsin-EDTA solution.

H295R cells (human adrenocortical carcinoma) were grown in 75-cm^2^ flasks in Dulbecco’s modified Eagle medium/HamF12 (DMEM/F12) containing HEPES buffer, l-glutamine, and pyridoxine HCl (#11330032, Thermo Fisher Scientific, Waltham, MA, USA). Additional supplements were added to the medium, including 0.00625 mg/mL insulin (#I9278, Sigma, St. Louis, MO, USA), 0,00625 mg/mL human transferrin (#T5391, Sigma, St. Louis, MO, USA), and 6.25 ng/mL selenous acid (#S9133, Sigma, St. Louis, MO, USA) 1.25 mg/mL bovine serum albumine (#A9647, Sigma, St. Louis, MO, USA), 2.5% nu-serum (Zenon Bio Kft. Szeged, Hungary), and 1% penicillin-streptomycin (#P0781, Sigma, St. Louis, MO, USA).

### 4.2. Sdhb Silencing Using Small Interfering RNA (siRNA)

PC12 cells were seeded in six-well plates for 24 h before transfection with two Silencer Select small interfering RNAs (siRNA A: Sequence (5′–3′: GAUUAAGAAUGAAAUCHAUtt, siRNA ID: #s151576; siRNA B: Sequence (5′–3′: GCAAAGUCUCGAAAAUAUAtt, siRNA ID: #s220846) (Ambion, Thermo Fisher Scientific, Waltham, MA, USA) targeting SDHB using RNAiMAX Reagent (Invitrogen, Thermo Fisher Scientific, Waltham, MA, USA) according to the manufacturer’s protocol. For negative control, cells cultured under identical conditions were transfected with non-targeting Silencer Select siRNA (Ambion by Life Technologies). Specific effect of siRNA against *Sdhb* was verified by Western blot analysis.

### 4.3. Protein Extraction and Western Blot

Total protein was extracted with M-Per reagent (#78503, Thermo Fisher Scientific, Waltham, MA, USA), according to the manufacturer’s instructions. Protein concentrations were determined by BCA Assay (Sigma, St. Louis, MO, USA). Total protein was separated by 10–15% SDS polyacrylamide gel electrophoresis, transferred to a PVDF membrane, and incubated overnight with primary antibody against SDHB (5 μg/mL; anti-SDHB, ab14714, Abcam, Cambridge, United Kingdom). Spectra Multicolor Broad Range Protein Ladder (#26634, Thermo Fisher Scientific, Waltham, MA, USA) was used as a protein ladder. For loading control membranes were stripped and re-probed using mouse anti-β-actin (1:25,000, Cell Signaling Technology, ZA, Leiden, The Netherlands). Anti-mouse HRP-conjugated IgG was used as secondary antibody (1:2,000, #P044701, Agilent, Santa Clara, CA, USA). Band intensities were quantified using Image J software (National Institutes of Health, Bethesda, MD, USA).

### 4.4. Biochemical Inhibition of SDH Enzyme

Itaconic acid was purchased from Sigma (#I29204, Sigma-Aldrich (Sigma, St. Louis, MO, USA). The 500 mM stock solutions were prepared with nuclease free water; pH 7.2 was adjusted with NaOH.

Cells were seeded onto six-well plates. After 24 h incubation, the used medium was replaced by fresh medium, after washing with PBS. Then, 25 mM itaconic acid was added in the wells. Nuclease free water was used as control.

Atpenin A5 (atpenin) used for our study was a generous gift from Christos Chinopoulos. Atpenin was purchased from Enzo Life Sciences (#ALX-380-313-MC25, Enzo Life Sciences, Inc., Farmingdale, NY, USA). First, 2mM stock solution was prepared with absolute ethanol. Cells were seeded onto six-well plates. After 24 h incubation, the used medium was replaced by fresh medium, after washing with PBS. Then, 1 µM itaconic acid was added in the wells. Absolute ethanol in the same treatment volume was used as control.

### 4.5. Inhibition of GLS-1 Activity

Bis-2-(5-phenylacetamido-1,3,4-thiadiazol-2-yl) ethyl sulfide (BPTES) was purchased from Sigma (#SML0601, Sigma-Aldrich (Sigma, St. Louis, MO, USA). The 2mM stock solutions were prepared with DMSO. Then, 10 μM BPTES was added to the cells. DMSO was used as control.

### 4.6. Cell Viability and Proliferation Assays

AlamarBlue test was used (Thermo Fisher Scientific, Waltham, MA, USA) to determine the viability effects of itaconic acid and atpenin treatment after 24, 48, and 72 h in PC12, HeLa, and H295R cells and in PC12 cells after transfection of siRNA against *Sdhb* or mock siRNA and after co-treatment with BPTES. The assay was performed in 96-well plates. All treatments at each time point and siRNA transfections were performed in six replicates, outliers were excluded before the statistical analysis. For studying the viability changes with AlamarBlue assay, PC12 cells were plated in 100 µL cell culture media at a density of 5000 cells/well for 24h treatment, 2500 cells/well for 48 h treatment, and 1700 cells/well for 72 h treatment. HeLa cells were plated in 100 µL cell culture media at a density of 3000 cells/well for 24 h treatment, 1500 cells/well for 48 h treatment, and 1000 cells/well for 72 h treatment. H295R cells were plated onto 96-well culture plates in 100 uL cell culture media at a density of 10000 cells/well for 24 h treatment; 5000 cells/well for 48 h treatment; 3500 cells/well for 72 h treatment. After 24 h, cell media was replaced by fresh media, and itaconate, atpenin, or siRNA against *Sdhb* treatment was performed. After the given incubation time, 10 µL AlamarBlue, was added to each well. After 1 h and 15 min incubation at 37 °C, fluorescence was measured in the 560-590 nm range using Varioskan Flash plate reader (Thermo Fisher Scientific). Percentage of the cell proliferation was given relative to control samples.

Sulforhodamine B (SRB) assay was used for evaluation of proliferation of PC12 cells. The cells were seeded onto 96-well plates at a density of 2500 cells/well. Each measurement was performed six replicates. After incubation with the indicated drug concentrations for 24/48/72 h, cells were fixed by cold 10% trichloroacetic acid for 60 min in 4 °C, washed with water, and dried. After drying, cells were incubated with 0.4% sulforhodamine B (Sigma-Aldrich) for 15 min in RT. After washing with 1% acetic acid, the protein-bound dye was dissolved in 10 mM Tris. The absorbance at 570 nm was measured in LabSystems Multiskan RC/MS/EX Microplate Reader (Artisan Scientific, Champaign, IL, USA).

### 4.7. 3D Culturing of PC12 Cells

The PC12 rat cell line was seeded with a density of 500,000 cells per six-well (2 mL/well) at 37 °C and 5% CO_2_. For spheroid induction serum-free defined media (Lichner et al. 2015) containing Ham’s F-12K (Kaighn’s) Medium (Gibco; Thermo Fisher Scientific) with 2% B27 Supplement, 50 ng/mL EGF and 50 ng/mL FGF was used. After spheroid formation (96 h) cells were treated with 500 nM itaconate and 10 μM itaconate-BPTES solution for 24, 48, and 72 h.

In 3D structure biochemical assays for proliferation and viability are not reliable due to uncertain diffusion of the reagent into the inner/central part of the spheroids. Therefore, viable and dead cells were investigated by trypan blue staining method. Spheres were dissociated with trypsine then they were stained with 0.4% (*w/v*) trypan blue solution (Life Technologies, California, CA, USA). Cell growth and the number of live and dead cells were assessed under Burker chamber.

### 4.8. Measurement of the Intracellular Concentration of Metabolites Using Liquid Chromatography Mass Spectrometry (LC-MSMS)

Cells were grown in six-well plates. All experiments (treatment with itaconic acid and siRNA transfection) were made in three replicates except for 24 h itaconic acid treatment of PC12 cells, where nine biological replicates were carried out.

Intracellular metabolites (lactate, pyruvate, citrate, α-ketoglutarate, succinate, fumarate, malate, glutamate, aspartate) were extracted by a modified method based on Szoboszlai et al. [72]. In brief, the cells were quenched in liquid nitrogen and extracted by mixture of MeOH–chloroform–H_2_O (9:1:1) and vortexed at 4 °C. After centrifugation (15,000× *g*, 10 min, 4 °C) the clear supernatants were kept at −80 °C until liquid chromatography-mass spectrometry (LC-MS) measurements. The concentrations of lactate, citrate, succinate, fumarate, malate, glutamate, and aspartate were assessed by using calibration curves obtained with the dilution of analytical grade standards in the range of 0.5–50 µM. LC-MS assays were used by Perkin-Elmer Flexar FX10 ultra-performance liquid chromatograph coupled with a Sciex 5500 QTRAP mass spectrometer. Chromatographic separation was carried out on a Phenomenex Luna Omega C18 column (100 × 2.1 mm, 1.6 µm) (GenLab Ltd., Budapest, Hungary). The mobile phase consisted of water and methanol containing 0.1% (*v/v*) formic acid. The MS was operating in negative electrospray ionization mode. For the measurements the following settings were adjusted—source temperature: 300 °C ionization voltage: -4000 V, entrance potential: −10 V, curtain gas: 35 psi, gas1: 35 psi, gas2: 35 psi, CAD gas: medium. Multiple reaction monitoring (MRM) mode was applied to perform quantitative analyses. All samples were measured in triplicate. Concentrations of metabolites were normalized to DNA concentration isolated from cells plated, incubated, and treated in the same manner as cells used for metabolite analysis. The cells were trypsinized and DNA was extracted using the semiautomatic DNA isolation protocol with QIAcube instrument (Qiagen, Hilden, Germany). The concentration of the extracted DNA samples was measured with NanoDrop 1000 Spectrophotometer (Thermo Fisher Scientific, Waltham, MA, USA).

### 4.9. Expression of SDHB and Glutaminase Type 1 (GLS-1) in Hereditary Pheochromocytoma/Paraganglioma Tissues Using Immunohistochemistry

Representative tissue blocks (*n* = 35) from 29 patients with Pheo/PGL were evaluated by two expert pathologists. Eleven patients had hereditary Pheo/PGLs (five carried *SDHB* and six patients carried the *RET* mutation (Table 1). Informed consent was obtained from all subjects and the experiments conformed to the principles set out in the WMA Declaration of Helsinki. Our study was approved by the Scientific and Research Committee of the Medical Research Council of Ministry of Health, Hungary (ETT-TUKEB 4457/2012/EKU).

Malignancy was diagnosed when a tumor was recurring, or local or distal metastases were detected. Of 35 tissue samples 13 were classified as malignant (six related to *SDHB* while four to *RET* mutations, no pathogenic mutations were detected in three malignant cases). Immunostaining of SDHB and GLS-1 was performed as previously described [73]. In brief, 4 µm-thick sections of formalin-fixed paraffin-embedded were used. After deparaffinization and blocking the endogenous peroxidases, antigen retrieval was performed for 30 min (10 mM citrate pH 6.0) using a pressure cooker. Slides were incubated with anti-glutaminase (ab156876, Abcam) and anti-SDHB (ab14714, Abcam) primary antibodies. Immunohistochemical reactions were visualized using Novolink Polymer (Leica Biosystems, Wetzlar, Germany) detection system and 3,3′-Diaminobenzidine (DAB, Dako) chromogen, followed by hematoxylin counterstain. Immunoreactivity was assessed in tumor cells (and normal adrenal medulla cells as control) using H-scores [74], which range from 0 to 300 and were calculated by multiplying the intensity of staining (0—no staining, 1+—weak staining, 2+—moderate staining, or 3+—strong staining) and the percentage of immunopositive cells (0–100). For example, 40% of tumor cells staining positive with moderate intensity (2+) and 10% of the tumor cells staining with strong intensity (3+) results in an H-score of 110. Based on H-score, expression of SDHB and GLS-1 was classified as ‘low’ (H-score < 100) and ‘high’ (H-score ≥ 100).

### 4.10. GLS-1 Gene Expression Measurements

Experiments were performed in six-well plates in duplicates for RNA isolation. Total RNA was harvested using RNeasy Mini Kit (50) (#74104, Qiagen, Hilden, Germany), according to the manufacturer’s instructions. RNA concentrations were determined with NanoDrop 1000 Spectrophotometer (Thermo Fisher Scientific, Waltham, MA, USA). For the quantitative real-time PCR (qRT-PCR) experiments, 1 µg of total RNA was reverse transcribed using High-Capacity RNA-to cDNA Kit (#4387406, Thermo Fisher Scientific, Waltham, MA, USA), according to the manufacturer’s instructions. For gene expression measurements, predesigned TaqMan Gene Expression assays were used (rat GLS: Rn00561285_m1, human GLS-1: Rn00667869_m1, rat actin: Rn00667869_m1, human actin: Hs99999903_m1; all from Applied Biosystems by Life Technologies). cDNA was diluted 100×. All measurements were performed in triplicate. DeltaCT (dCT) values were calculated and deltadeltaCT (ddCT) values were normalized to the controls in the experiments. Fold change values were calculated from 2^−ddCT^.

### 4.11. Cellular Respiration

Seahorse XF96 Analyzer (Agilent Technologies, Santa Clara, CA, USA) was used to assess real-time oxygen consumption rate (OCR), reflecting mitochondrial oxidation and extracellular acidification rate (ECAR), based on previous descriptions [40,75,76]. PC12 cells were plated in 100 μL complete medium at 30,000 cells/well density onto 96-well Seahorse plates (Agilent Technologies, Santa Clara, CA, USA) 24 h prior to the assays. Itaconate (25 mM) or atpenin (1 μM) treatment was carried out 24 or 48 h before the assays, whereas transfection with siRNA against *Sdhb* or mock vector 48 h before the assays. BPTES treatment was carried out 24 h prior to the assay. On the day of the assay complete medium was removed and was replaced by a medium containing (in mM): 120 NaCl, 3.5 KCl, 1.3 CaCl_2_, 1.0 MgCl_2_, 20 HEPES, 10 glucose at pH 7.4. The basal OCR and ECAR values were calculated after 1.5 h incubation at this condition.

During the measurements freshly prepared glutamine (4 mM) and/or metabolic inhibitors/modulators (oligomycin 2 μM, 2,4-dinitrophenol- DNP 200 μM and antimycin A + rotetone 1-1μM) were injected into each well to reach the desired final working concentration.

### 4.12. Oxygen Consumption of PC12 Cells

Oxygen consumption was performed polarographically using an Oxygraph-2k (Oroboros Instruments, Innsbruck, Austria). Two T75 flasks of approx. 80% confluent PC12 cells were suspended in 2 mL incubation medium, containing, in mM: mannitol 225, sucrose 125, Hepes 5, EGTA 0.1, KH_2_PO_4_ 10, MgCl_2_ 1, glutamate 5, malate 5, succinate 5, 0.5 mg/mL bovine serum albumin (fatty acid-free), pH  =  7.25 (KOH). Experiments were performed at 37 °C in 8–12 parallel wells. Oxygen concentration and oxygen flux (pmol·s^−1^·mg^−1^; negative time derivative of oxygen concentration, divided by mitochondrial mass per volume) were recorded using DatLab software (Oroboros Instruments).

### 4.13. SDH Activity Measurement

SDH activity was assessed as described previously [77]. Briefly, the oxidation of succinate by decylubiquinone was coupled to the reduction of dichlorophenolindophenol (DCPIP), and the rate was followed spectrophotometrically at 600 nm at 30 °C.

### 4.14. Statistical Analysis

All data are expressed as mean ±SD except where it is indicated otherwise. Statistical analysis was performed using GraphPad Prism 6 software (GraphPad Software Inc., La Jolla California, CA, USA). Gaussian distribution of data was evaluated with Shapiro–Wilks test. In the case of normally distributed data the differences were analyzed by Student’s *t*-test, otherwise by rank sum test. Correlation in case of normally distributed data was calculated with a Pearson test, otherwise a Spearmen test was used. *p* values of <0.05 were considered to be statistically significant.

## 5. Conclusions

In summary, we demonstrated for the first time that SDH inhibition either with itaconate, atpenin, or *SDHB* knockdown had a positive effect on cell viability of chromaffin cells but not on other cell lines which may be related to the glutamine/glutamate metabolism. The aim of our study was to establish a cost-efficient model for the research of novel prognostic factors and therapeutic agents before conducting further, more complex and more expensive studies, however we acknowledge the limitations of our research. Lack of availability of SDHB-mutant animal model developing Pheos [20] warrants other in vitro and in vivo models for deciphering the mechanism contributing to the malignant behavior of these rare tumors. SDHB expression in some *SDHB*-mutant Pheo/PGL tissues suggests that tumor heterogeneity occurs even in *SDHB*-associated tumors. In addition, by measuring the succinate to fumarate ration in various Pheo tumors it was shown that the remaining SDH activity was highly variable [63]. All together these data suggest that some SDH activity is still maintained in these tumors, therefore knockdown of *SDHB* by siRNA provides a feasible model for the disease. As itaconate treatment of PC12 cells successfully mimicked the phenotype observed in the *Sdhb* silenced cells, it can be a useful, easily accessible in vitro model for these tumors. The importance of glutamine/glutamate metabolism of cells lacking SDH was confirmed by our in vitro experiments demonstrating the upregulation of GLS-1 after SDH inhibition (either by chemical agents or *Sdhb* knockdown) and by the decreased proliferation upon GLS-1 inhibition. The importance of GLS-1 was also reassured by evaluation of expression of GLS-1 in malignant PGL tissues compared to benign tumors. Our data suggests that GLS-1 inhibition in SDH deficient chromaffin cells tumors may represent novel, tumor specific alternatives of therapy in malignant Pheo/PGL where the current treatment options are limited. Moreover, as reliable markers of malignant Pheos are lacking, GLS-1 staining seems to be worthy of further investigations as a potential marker of Pheo/PGL malignancy.

## Figures and Tables

**Figure 1 cancers-12-00599-f001:**
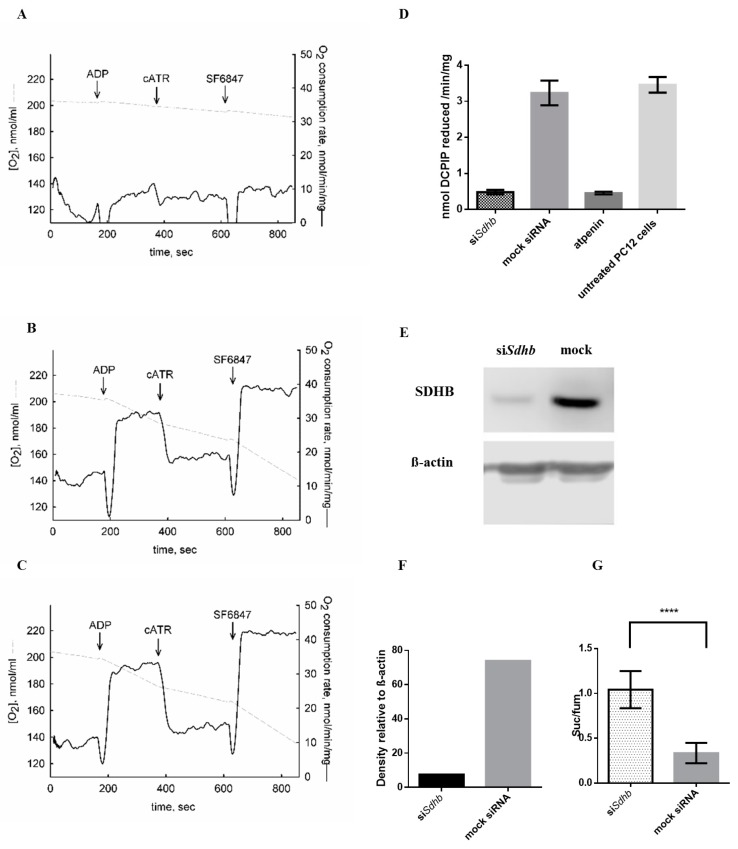
Effects of *Sdhb* knockdown in PC12 cells. (**A**–**C**) Oxygen consumption PC12 cells. Grey trace represents the negative time derivative of oxygen concentration, divided by mitochondrial mass per volume. Additions of substances are indicated by arrows. ADP: 0.2 mM. cATR: 2 µM. SF 6847: 1 µM. (**A**) Respiration of si*Sdhb* transfected PC12 cells. (**B**) Respiration of mock transfected PC12 cells. (**C**) Respiration of untreated PC12 cells. (**D**) Succinate dehydrogenase (SDH) activity after si*Sdhb* transfection, mock siRNA transfection, atpenin treatment and in untreated PC12 cells. The effectiveness of si*Sdhb* transfection was compared to atpenin, which is a well-known potent SDH inhibitor. (**E**) SDHB protein level after siRNA transfection using Western blot analysis. (**F**) Densitometry quantification of the SDHB protein in si*RNA* against *Sdhb* treated and mock siRNA transfected PC12 cells. (**G**) Succinate to fumarate ratio in PC12 cells transfected with *SDHB* targeting siRNA (si*Sdhb*) compared to mock transfected cells. suc/fum: succinate to fumarate ratio. ****: *p* < 0.0001.

**Figure 2 cancers-12-00599-f002:**
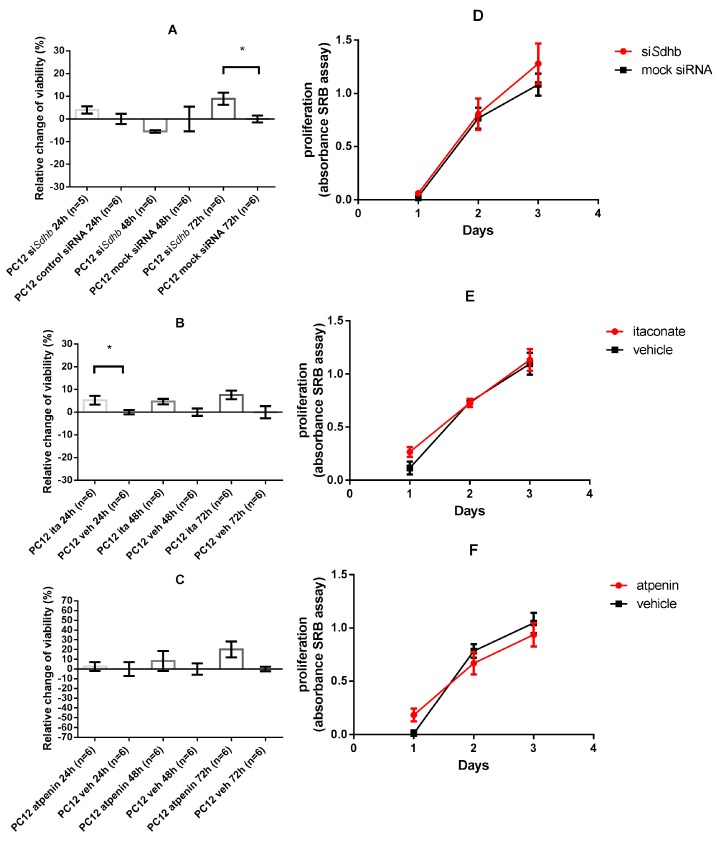
Cell viability and proliferation of PC12 cell lines. Data is presented in mean ± SEM. Cell viability measurement values are normalized to the control values. A total of 100% was subtracted from the values, therefore, changes in % compared to control are presented. (**A**–**C**) Relative change of cell viability of *Sdhb* targeting siRNA transfected, itaconate, or atpenin treated PC12 cells after 24, 48, and 72 h incubation, compared to control. (**D**–**F**) Proliferation of PC12 cells after *Sdhb* targeting siRNA transfection, itaconate, or atpenin treatment measured by SRB assay. Ita: Itaconate; veh: vehicle. SRB: Sulforhodamine B; *: *p* < 0.05.

**Figure 3 cancers-12-00599-f003:**
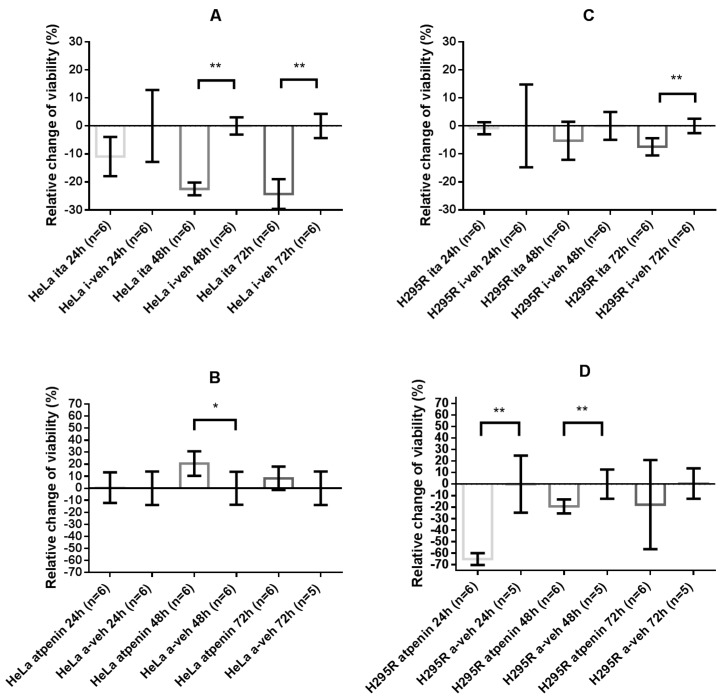
Cell viability measurements of HeLa and H295R cell lines. Values are normalized to the control values. A total of 100% was subtracted from the values, therefore, changes in % are presented compared to control. (**A**) Relative change of cell viability of HeLa cells after 24, 48, and 72 h itaconate treatment, compared to control. (**B**) Relative change of cell viability of HeLa cells after 24, 48, and 72 h atpenin treatment, compared to control. (**C**) Relative change of cell viability of H295R cells after 24, 48, and 72 h itaconate treatment, compared to control. (**D**) Relative change of cell viability of H295R cells after 24, 48, and 72 h atpenin treatment, compared to control. Veh: vehicle. *: *p* < 0.05; **: *p* < 0.01.

**Figure 4 cancers-12-00599-f004:**
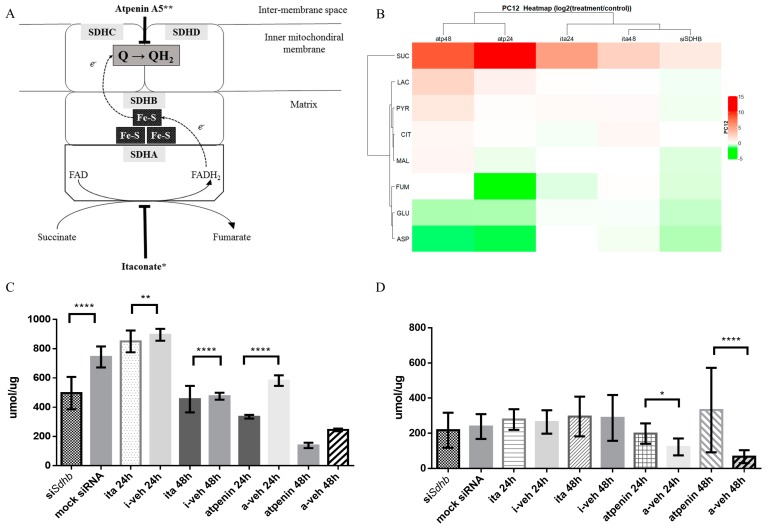
Inhibitory method specific changes in cellular metabolomics of PC12 cells after SDH inhibition. (**A**) Schematic illustration of Complex II. FAD: flavin-adenin-dinucleotide; FADH2: reduced FAD; Fe-S: iron-sulfur cluster; Q: coenzyme-Q or Ubiquinone; QH2: ubiquinol; SDHA: succinate dehydrogenase subunit A; SDHB: succinate dehydrogenase subunit B; SDHC: succinate dehydrogenase subunit C; SDHD: succinate dehydrogenase subunit D. *: the presumed inhibitory activity of itaconic acid. **: The presumed inhibitory activity of atpenin A5. (**B**) HeatMap: visualization of changes in metabolite concentrations in PC12 cell line after *Sdhb* knockdown, itaconate and atpenin treatment. Fold-changes of different metabolic concentration were calculated (values measured after treatment were divided with control values) then the given values were log2 transformed. These values were used for construction of the heatmap and represented with color scale (red+/white 0/green−). (**C**) Normalized glutamate concentrations in PC12 cells after *Sdhb* knockdown, itaconate and atpenin treatment, and vehicle treatments. (**D**) Normalized lactate concentrations in PC12 cells after *Sdhb* knockdown, itaconate and atpenin treatment, and vehicle treatments. atp24: 24 h atpenin treatment; atp48: 48 h atpenin treatment; ita24: 24 h itaconate treatment; ita48: 48 h itaconate treatment; siSDHB: *Sdhb* knockdown; SUC: succinate; LAC: lactate; PYR: pyruvate; CIT: citrate; MAL: malate; FUM: fumarate; GLU: glutamate; ASP: aspartate. *: *p* < 0.05; **: *p* < 0.01; ****: *p* < 0.0001.

**Figure 5 cancers-12-00599-f005:**
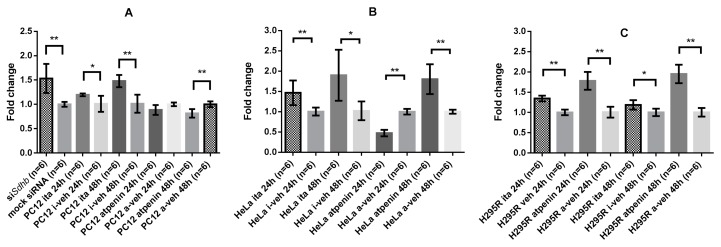
Glutaminase-1 (GLS-1) expression of PC12 and HeLa cells upon SDH inhibition: all values are normalized to control. (**A**) Fold changes in GLS-1 expression in PC12 cells upon *Sdhb* knockdown, itaconate treatment, and atpenin treatment. (**B**) Fold changes in GLS-1 expression in HeLa cells upon itaconate and atpenin treatment. (**C**) Fold changes in GLS-1 expression in H295R cells upon itaconate and atpenin treatment. Ita: itaconate; i-veh: control for itaconate experiments; a-veh: control for atpenin experiments; *: *p* < 0.05; **: *p* < 0.01.

**Figure 6 cancers-12-00599-f006:**
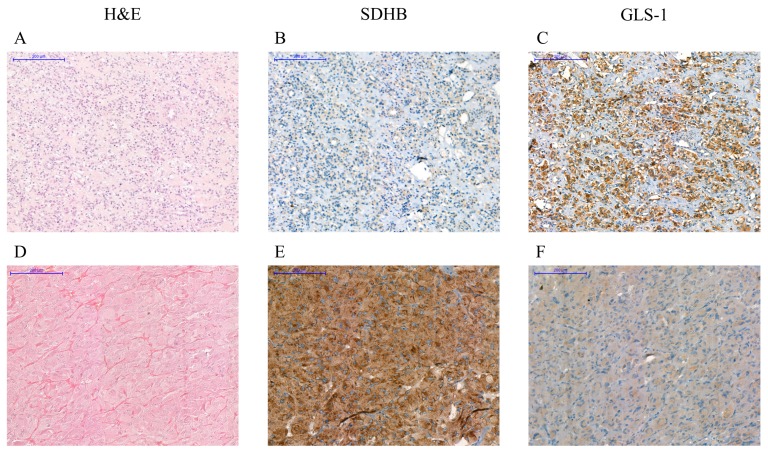
Immunohistochemistry: Immunostaining with antibodies against SDHB and GLS-1 of a paraganglioma associated with *SDHB* p. mutation (**A**–**C**) and a *RET* p.C634W-associated pheochromocytoma (**D**–**F**). Lack of SDHB staining in *SDHB* mutated tumors (**B**) and strong GLS-1 signal was detected in malignant *SDHB*-associated tumor (**C**). Lack of GLS-1 positive cells can be observed in *RET*-associated benign pheochromocytoma (**F**). Scale bar = 200 µm. H&E: hematoxylin and eosin staining. SDHB: succinate dehydrogenase subunit B staining. GLS-1: glutaminase-1 staining.

**Figure 7 cancers-12-00599-f007:**
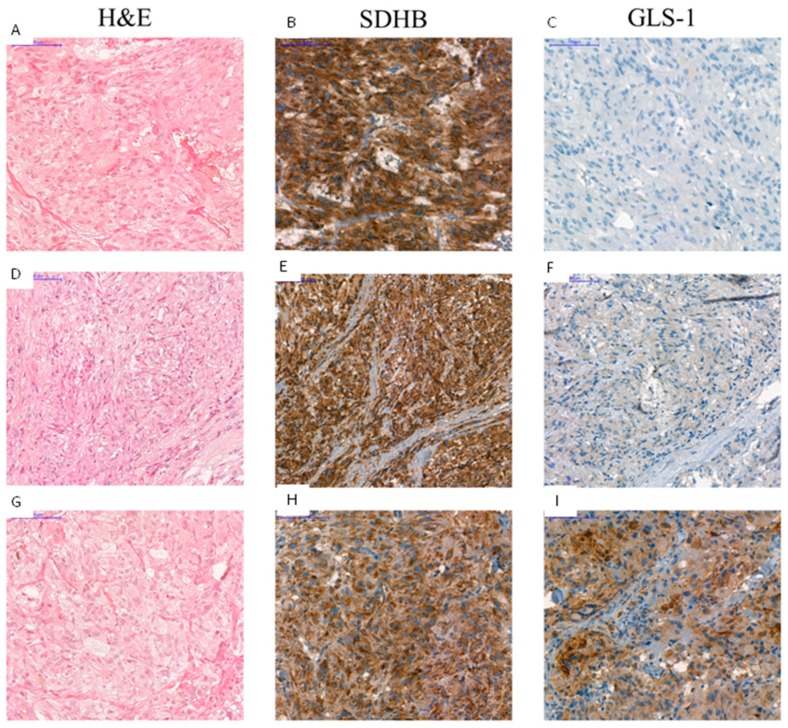
Immunostaining with antibodies against SDHB and GLS-1 of a *RET* p.C634R-associated pheochromocytoma. Panel **A**, **D**, **G** shows the hematoxylin and eosin stains of the primary tumor (**A**), the first reoccurring tumor (**D**) and the second reoccurring tumor (**G**). Strong SDHB staining (**B**,**E**,**H**) and lack of GLS-1 positive cells can be observed in a primary benign tumor (**C**). Slightly visible GLS-1 staining can be observed in the first reoccurring, invasive tumor (**F**) while strong GLS-1 signal was detected in the second reoccurring, invasive tumor (**I**). Scale bar = 100 µm. H&E: hematoxylin and eosin staining. SDHB: succinate dehydrogenase subunit B staining. GLS-1: glutaminase-1 staining.

**Figure 8 cancers-12-00599-f008:**
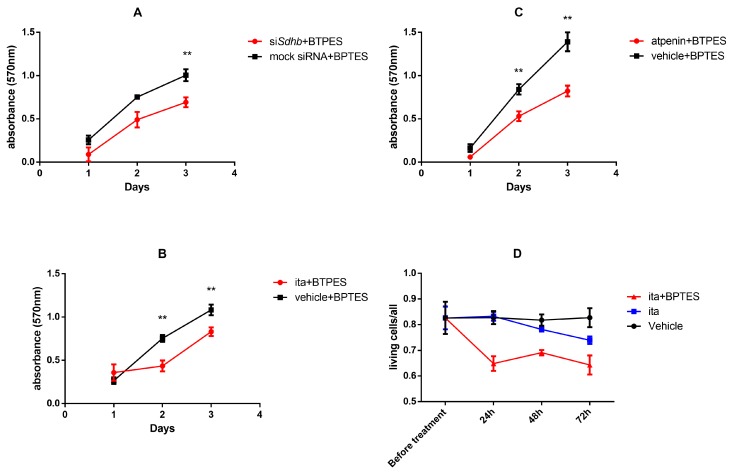
Proliferation of SDH impaired PC12 cells upon BPTES treatment. Cell proliferation was assessed by SRB assay. (**A**–**C**) Effects of SDH impairment and BPTES treatment on cell proliferation of PC12 cells cultured in monolayer. Experiments were performed in hexuplicates. (**D**) The effects of itaconate and BPTES treatment in PC12 spheroid cell culture. The ratio of living cells to total number of cells is shown before treatment and after 24, 48, and 72 h incubation. All experiments were performed at least six times in each group. Values are shown as mean ± standard error mean. SRB: Sulforhodamine B; BPTES: bis-2-(5-phenylacetamido-1,2,4-thiadiazol-2-yl)ethyl sulfide; ITA: itaconate; **: *p* < 0.01.

**Figure 9 cancers-12-00599-f009:**
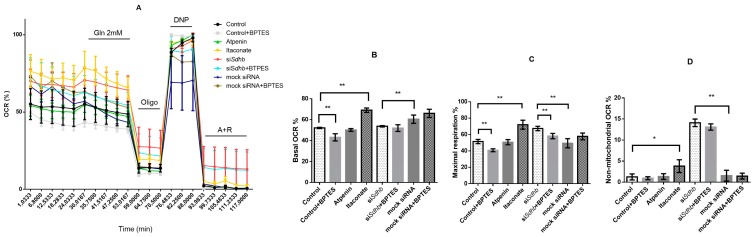
Oxygen consumption measurements of PC12 cells. (**A**). Oxygen consumption ratio (OCR %) of PC12 cells. The minimum value of OCR is 0%, maximum is 100%. (**B**) Basal respiration: subtraction of non-mitochondrial respiration from the baseline respiration. (**C**). Maximal respiration of PC12 cells: difference of OCR after 2,4-dinitrophenol (DNP) and after antimycin A + rotetone (A + R) admission. (**D**) Non-mitochondrial respiration: OCR after inhibition of both complexes I and III with A + R. OCR: oxygen consumption ratio; Gln: glutamine; Oligo: oligomycin; DNP: 2,4-dinitrophenol; A + R: antimycin A + rotetone. *: *p* < 0.05; **: *p* < 0.01.

**Table 1 cancers-12-00599-t001:** Immunohistochemical evaluation of expression of SDHB and GLS-1 in Pheo/PGL tumor samples.

Nr.	Type of Tumor Tissue	Biological Behavior	Germline Mutation	Age at Surgery Years	SDHB	GLS-1
					H-Score	H-Score
1	PGL	Malignant	*SDHB* p.C243Y	32	0	15
2	PGL	Malignant		34	65	150
3	PGL	Malignant	*SDHB* p.C196G	32	10	160
4	Pheo	Malignant *	*SDHB* p.T88I and R90 frame shift	14	90	110
5	Pheo			15	70	130
6	Pheo	Malignant Benign	*RET* p.C634R	18	160	0
7	Pheo	Malignant		21	155	5
8	Pheo	Malignant		22	210	10
9	Pheo	Malignant		25	150	110
10	Pheo	Benign	*RET* p.C634W	31	123	37
11	Pheo	Malignant		34	190	160
12	Pheo	Benign (bilateral)	*RET* p.C634R	46	200	240
13	Pheo	Benign	*RET* p.C634Y	34	115	20
14	Pheo	Benign	*RET* p.C609S	42	100	10
15	Pheo	Benign	*RET* p.C634Y	63	157	7
16	Pheo	Benign	sporadic	49	110	10
17	Pheo	Malignant	sporadic	56	80	20
18	Pheo	Benign	sporadic	47	140	120
19	Pheo	Benign	sporadic	27	30	55
20	PGL	Malignant	*SDHB* c.424-1G>A	54	0	40
21	Pheo	Benign	sporadic	62	180	95
22	PGL	Malignant	sporadic	82	160	10
23	Pheo	Malignant	sporadic	18	120	10
24	Pheo	Benign	sporadic	55	90	105
25	Pheo	Benign	sporadic	56	110	10
26	PGL	Benign	sporadic	30	10	135
27	Pheo	Benign	sporadic	41	115	10
28	Pheo	Benign	sporadic	79	110	5
29	Pheo	Benign	sporadic	53	200	105
30	Pheo	Benign	sporadic	43	180	20
31	Pheo	Benign	sporadic	71	-	90
32	Pheo	Benign	*SDHB* p.Q109X	47	10	10
33	Pheo	Benign	sporadic	54	190	20
34	Pheo	Benign	sporadic	65	140	30
35	Pheo	Benign	sporadic	59	90	115

Tumors were considered malignant when a tumor was recurrent or local/distant metastasis were documented. * the patient was presented with a 16 × 13 × 9 cm Pheo, histology did not approve malignancy, but preoperative MRI described multiple bone metastases. n.a.: not available; Pheo: pheochromocytoma; PGL: paraganglioma.

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
