# Peer review of "Glutaminases as a Novel Target for SDHB-Associated Pheochromocytomas/Paragangliomas"

_cancers, 2020, doi:10.3390/cancers12030599_

Round 1
Reviewer 1 Report
The authors evaluate whether glutaminases can be a new opportunity to target SDHB mutated pheochromocytomas/paraganglionas (PPGLs). This is an interesting area of research, which deserves more investigation.
Comments:
- When evaluating the metabolomics changes, why is it that the PC-12 cells had less rise in succinate compared to the other cell lines such as HeLa and H295R cells?
- In the text it mentions in the Results, that intracellular glutamate increases in H295R cells as a result of itaconic acid treatment. These results do not seem to correlate with the heatmap presented.
- In Figure 5, the authors may want to comment on how the HeLa and H295R cells responded more consistently to the itaconate and atpenin treatments in terms of GLS-1 expression compared to PC-12 cells. Is it because of their malignant nature?
- Could the authors speculate why in Table 1 that patients 12, 18, 24, 26, 29, and 35 have a high GLS-1 H-score as these tumors are benign?
- In the Discussion, the authors may want to comment on why the malignant RET tumors would increase GLS-1 expression. How would this observation correlate with an SDHB mutation?
- In Figure 6, please label the panels with the appropriate labels, “A,B,C,D, etc.”
- The authors need to place the correct legend with the correct Figures in the Supplementary files.
- The authors need to review the article carefully for spelling and punctuation errors; a few examples:
- Introduction: “(incidence: 0,8 per)”
- Figure 4: spelling: “coenzime; panle”
- Discussion: “piruvate”
- Conclusion” “lutamine/glutamate”
Author Response
Reviewer 1
The authors evaluate whether glutaminases can be a new opportunity to target SDHB mutated pheochromocytomas/paraganglionas (PPGLs). This is an interesting area of research, which deserves more investigation.
Answer: we thank the Reviewer for this positive comment.
Comments:
- When evaluating the metabolomics changes, why is it that the PC-12 cells had less rise in succinate compared to the other cell lines such as HeLa and H295R cells?
Answer: We thank the Reviewer for this question. As it can be seen in Supplementary Tables 1 and 2, the normalized concentrations of succinate measured in PC12 cells are between those measured in HeLa and H295R cells depending on SDH inhibition method. These data suggest again that the metabolomics of various cancer cells are highly variable and it highlights again that adequate in vitro models are needed for accurate characterization of each tumor cell type. We can only speculate that PC12 cells might direct the metabolites away from the Kreb’s cycle more effectively (e.g. reductive metabolism of glutamine), therefore the “succinate trap” is not as significant as it is HeLa and H295R cells.
- In the text it mentions in the Results, that intracellular glutamate increases in H295R cells as a result of itaconic acid treatment. These results do not seem to correlate with the heatmap presented.
Answer: We thank the Reviewer for drawing our attention to this discrepancy between heatmap and concentration measured. Based on this comment together with other suggestions we decided to remove the heatmaps about metabolomics of HeLa and H295R cells from the main text and present them as Supplement. However, as it can be seen in Supplementary Table 1 and 2, the glutamate concentration was increased in H295R cells after 24 h treatment but not after 48h treatment with itaconate (489,7±113,5 versus 328±63, p=0,0188). We think that the original color coding used on heatmaps has not allowed enough high resolution for presenting all changes, therefore our main findings regarding glutamate and lactate concentrations have been presented as column diagrams in our new Figure 4.
- In Figure 5, the authors may want to comment on how the HeLa and H295R cells responded more consistently to the itaconate and atpenin treatments in terms of GLS-1 expression compared to PC-12 cells. Is it because of their malignant nature?
Answer: We thank this important question. Based on our data presented on Figure 5, the expression of GLS-1 was very homogeneous across all these cell lines. However, we agree with the Reviewer that both itaconate and atpenin treatments caused very similar effect on HeLa and H295R cells. We agree with the Reviewer that behind this observation the malignant nature of these cells could stay. However, further studies using other anticancer therapies are needed for clarification of the role of GLS-1 in these cancer cells. We included one comment in our revised version.
- Could the authors speculate why in Table 1 that patients 12, 18, 24, 26, 29, and 35 have a high GLS-1 H-score as these tumors are benign?
Answer: We thank the Reviewer for this suggestion. In the revised version, we included a new paragraph about these observations “.It should be also mentioned that in some cases increased GLS-1 expression was not associated with malignancy. On the other hand, determination of malignancy in Pheo is difficult, because there is no obvious marker for it. Several studies, including a study published by Stenman et al., showed that even in RET-associated Pheos, using the PASS and GAPP algorithms, the malignancy was over-diagnosed [60] In our study we considered a Pheo malignant when the tumor was recurring or local or distal metastases were detected. In our in vitro experiments, the increased GLS-1 expression was not necessarily associated with increased viability, suggesting that for increased proliferation, other factors are also needed. The importance of GLS-1 may be the most important in SDH-compromised cells, where the concomitant inhibition of SDH and GLS-1 could result in cell lethality. The heterogeneous phenotype associated with Sdhb mutations is highlighted in an in vivo model of Sdhb mutation developed in C. elegans, where the deleted mutant arrested at in development, while the point mutant presented only infertility [71]. ”
- In the Discussion, the authors may want to comment on why the malignant RET tumors would increase GLS-1 expression. How would this observation correlate with an SDHB mutation?
Answer: Thank you for this suggestion, a comment was added to the discussion regarding this observation.
- In Figure 6, please label the panels with the appropriate labels, “A,B,C,D, etc.”
Answer: Corrections were made accordingly.
- The authors need to place the correct legend with the correct Figures in the Supplementary files.
Answer: Corrections were made accordingly.
- The authors need to review the article carefully for spelling and punctuation errors; a few examples:
- Introduction: “(incidenc e: 0,8 per)”
- Figure 4: spelling: “coenzime; panle”
- Discussion: “piruvate”
- Conclusion” “lutamine/glutamate”
Answer: These typos were corrected.
Finally, we would like to ask the Reviewer for his/her valuable comments which significantly contributed to improvement of our manuscript.
Reviewer 2 Report
In the manuscript titled "Glutaminases as a novel target for SDHB-associated pheochromocytomas/paragangliomas" prepared by Sarkadi et al., the author conducted an investigation on Sdhb knockdown PC12 cells. Although this study is limited by a lack of another cellular model as well as a preclinical animal study, this work provides useful information to SDHB depleted cells. I have several concerns as follow:
Several studies have reported CRISPR/Cas9 knock out cell line with SDHB depletion. Whereas in the present study an RNAi transient knockdown model is applied. The author should justify whether this is an ideal strategy. The present study lacks of preclinical animal models, which greatly limit the enthusiasm for a translational investigation. The author should perform or discuss the preclinical study to expand the scope of their findings. In Figure 4B, C and D, it seems like the metabolites were compared across cells treated with itaconic acid, atpenin, and siRNA, whereas the control group is not shown. I would suggest the author show each control, including non-treated and control RNA, in the same chart. The author showed a differential expression of GLS-1 in SDHB wild type and depleted tissue specimen. I would recommend performing image quantification and statistic analysis on GLS-1 expression. Currently, there is a pharmacological grade glutaminase inhibitor CB-839 in clinical investigation. The author may test their findings with this compound to provide more useful information for clinical translation. The author should mention several recent key publications regarding SDHB-depleted cancer. A few examples are enlisted here: PMID: 29636359, PMID: 26719882, PMID: 31979226, and PMID: 30013182Author Response
Comments and Suggestions for Authors
In the manuscript titled "Glutaminases as a novel target for SDHB-associated pheochromocytomas/paragangliomas" prepared by Sarkadi et al., the author conducted an investigation on Sdhb knockdown PC12 cells. Although this study is limited by a lack of another cellular model as well as a preclinical animal study, this work provides useful information to SDHB depleted cells. I have several concerns as follow:
- Several studies have reported CRISPR/Cas9 knock out cell line with SDHB depletion. Whereas in the present study an RNAi transient knockdown model is applied. The author should justify whether this is an ideal strategy.
Answer: We thank the Reviewer for his/her comment. When we started our work (5 years ago) we had no expertise in CRISP/Cas9 technology. However, we successfully used siRNA inhibition. As it can be seen on Figure 1, in PC12 cells this technology resulted in a 90% reduction of Sdhb level compared to mock transfected cells. In addition, knockdown of Sdhb with siRNA in PC12 rat chromaffin cells successfully inhibited SDH activity and increased succinate/fumarate ratio by >3 fold compared to mock siRNA transfected cells. This ratio is similar to those measured in tumor tissues for SDH mutant Pheo/PGLs. These explanations have been included in our manuscript.
- The present study lacks of preclinical animal models, which greatly limit the enthusiasm for a translational investigation. The author should perform or discuss the preclinical study to expand the scope of their findings.
Answer: We thank the Reviewer for this comment. A sentence in the Introduction section has been included about the lack of animal models developing pheochromocytoma due to SDHx mutations.
- In Figure 4B, C and D, it seems like the metabolites were compared across cells treated with itaconic acid, atpenin, and siRNA, whereas the control group is not shown. I would suggest the author show each control, including non-treated and control RNA, in the same chart.
Answer: we thank the Reviewer for this comment. We redraw the Figure and the main metabolic findings (glutamate and lactate) detected in chromaffin cells have been presented as column graphs. All concentrations measurements have been presented in Supplementary tables 1 and 2. The figure legend was corrected.
- The author showed a differential expression of GLS-1 in SDHB wild type and depleted tissue specimen. I would recommend performing image quantification and statistic analysis on GLS-1 expression.
Answer: We thank the Reviewer for this comment. GLS-1 and SDHB staining was quantified using H-scores, based on publication of Konosu-Fukaya et al. ( Reference nr. 74). This scoring system uses both intensity of staining and percentage of immunopositive cells. For details please see in Materials and Methods section. After scoring, statistical analysis have been made.
- Currently, there is a pharmacological grade glutaminase inhibitor CB-839 in clinical investigation. The author may test their findings with this compound to provide more useful information for clinical translation.
Answer: we thank the Reviewer for his/her suggestion. When we started our work this inhibitor was not available in our country. However, for inhibition of SDH we used three different approaches in order to investigate whether SDHB knockdown by siRNA results in a similar phenotype observed after pharmacological inhibition of SDH. Based on our metabolomics study, it turned out that GLS-1 might be a good candidate for inhibition of cell proliferation of SDH impaired cells. Therefore, we used BPTES a potent (13x more potent than CB-839), GLS-1 inhibitor in order to test whether successful inhibition of GLS-1 might be useful in inhibition of proliferation of this cells. We incorporated a comment in the revised version.
- The author should mention several recent key publications regarding SDHB-depleted cancer. A few examples are enlisted here: PMID: 29636359, PMID: 26719882, PMID: 31979226, and PMID: 30013182
Answer: We thank the Reviewer for these suggestions. All of them are incorporated in the revised version.
Finally, we would like to ask the Reviewer for his/her valuable comments which significantly contributed for improvement of our manuscript.
Round 2
Reviewer 2 Report
The author has addressed most of my concerns.